# The Reduction in Gastric Atrophy after *Helicobacter pylori* Eradication Is Reduced by Treatment with Inhibitors of Gastric Acid Secretion

**DOI:** 10.3390/ijms20081913

**Published:** 2019-04-18

**Authors:** Ryota Niikura, Yoku Hayakawa, Yoshihiro Hirata, Keiji Ogura, Mitsuhiro Fujishiro, Atsuo Yamada, Tetsuo Ushiku, Mitsuru Konishi, Masashi Fukayama, Kazuhiko Koike

**Affiliations:** 1Department of Gastroenterology, Graduate School of Medicine, The University of Tokyo, 7-3-1 Hongo, Bunkyo-ku, Tokyo 113-8655, Japan; HIRATAY-INT@h.u-tokyo.ac.jp (Y.H.); mtfujish-kkr@umin.ac.jp (M.F.); yamada-a@umin.ac.jp (A.Y.); konishi524326@gmail.com (M.K.); kkoike-tky@umin.ac.jp (K.K.); 2Department of Gastroenterology, Tokyo Metropolitan Police Hospital, Tokyo 164-8541, Japan; keiji-tky@umin.ac.jp; 3Department of Gastroenterology & Hepatology, Nagoya University Graduate School of Medicine, Nagoya 466-8560, Japan; 4Department of Pathology, Graduate School of Medicine, The University of Tokyo, Tokyo 113-8655, Japan; usikut@gmail.com (T.U.); mfukayama-tky@umin.ac.jp (M.F.)

**Keywords:** *H. pylori* eradication, atrophy, proton pump inhibitors (PPIs)

## Abstract

Background: *Helicobacter pylori* (*H. pylori*) eradication therapy may improve gastric atrophy and intestinal metaplasia, but the results of previous studies have not always been consistent. The aim of this study was to compare the histological changes of intestinal metaplasia and gastric atrophy among the use of acid-suppressing drugs after *H. pylori* eradication. Methods: A cohort of 242 patients who underwent successful eradication therapy for *H*. *pylori* gastritis and surveillance endoscopy examination from 1996 to 2015 was analyzed. Changes in the histological scores of intestinal metaplasia and atrophy according to drug use (proton-pump inhibitors (PPIs), H_2_ receptor antagonists (H2RAs), and non-acid suppressant use) were evaluated in biopsies of the antrum and corpus using a generalized linear mixed model in all patients. Results: The mean follow-up period and number of biopsies were 5.48 ± 4.69 years and 2.62 ± 1.67 times, respectively. Improvement in the atrophy scores of both the antrum (*p* = 0.042) and corpus (*p* = 0.020) were significantly superior in patients with non-acid suppressant drug use compared with those of PPI and H2RA use. Metaplasia scores in both the antrum and corpus did not improve in all groups, and no significant differences were observed among groups in the antrum (*p* = 0.271) and corpus (*p* = 0.077). Conclusions: Prolonged acid suppression by PPIs or H2RAs may limit the recovery of gastric atrophy following *H. pylori* eradication.

## 1. Introduction

*H*. *pylori* eradication therapy may have the potential to regress atrophic and metaplastic changes and subsequently prevent the development of gastric cancer. In fact, previous studies have suggested that *H*. *pylori* eradication reduces metachronous recurrence of gastric cancer after endoscopic surgery [1,2,3,4,5]. It has been suggested that *H*. *pylori* eradication partly improves atrophy and metaplasia as well [6], although there are numerous reports which failed to demonstrate such histological reversion, and indeed the improvement of metaplasia after eradication has not been confirmed by meta-analyses.

Previous studies have reported that oxyntic atrophy seems to be a pathogenic factor in gastric carcinogenesis [7,8,9]. Reduced gastric acidity in the atrophic stomach leads to hypergastrinemia, which stimulates the function and proliferation of enterochromaffin-like cells located in the oxyntic mucosa, and potentially promotes gastric carcinogenesis in the oxyntic mucosa [10]. Recently, we and others have reported that long-term use of proton pump inhibitors (PPIs) may be a risk factor for gastric cancer after *H. pylori* eradication [11,12]. It is well known that acid suppression by PPIs and other drugs causes hypergastrinemia via a negative feedback mechanism, which may be associated with the subsequent risk of gastric cancer [10]. In contrast, the role of metaplasia in gastric carcinogenesis has been questioned in recent years, and metaplasia may just be a marker of long-term atrophic gastritis [8,9,13,14]. 

In this report, we conducted a retrospective cohort study which collected clinical and histological data from *H*. *pylori*-eradicated patients. We quantitatively evaluated the reversibility of histological findings of intestinal metaplasia, atrophy, and inflammation using the updated Sydney system, and investigated the associations with several clinical and molecular parameters, focusing especially on the use of acid-suppressing drugs.

## 2. Results

### 2.1. Patients

A total of 242 eligible patients who underwent successful eradication therapy for *H*. *pylori* gastritis were analyzed (Figure 1A, study flow chart). The baseline characteristics of the patients by drug use are shown in Table 1. PPI, H_2_ receptor antagonists (H2RA), and non-acid suppressant drug use were 110 (45.45%), 33 (13.64%), and 99 (40.91%) patients, respectively. The mean durations of PPI and H2RA use were 915 days and 838 days, respectively. No significant differences in baseline characteristics were observed among drug users, except for non-steroidal anti-inflammatory drug (NSAID) use (*p* < 0.001). The mean follow-up period was 5.48 ± 4.69 years.

### 2.2. Changes in the Updated Sydney System Scores and Association with Drug Use

Of the 242 patients, the metaplastic histological scores improved in 36 patients and were exacerbated in 27 patients.

Associations between changes in the updated Sydney system scores and drug use are shown in Table 2. Improvements in the atrophy scores of both the antrum and corpus were significantly superior in patients with non-acid suppressant use compared with those of PPI and H2RA users (*p* = 0.042, 0.020). Metaplasia scores did not improve in either drug users or non-users, and no significant differences were observed among the groups. Neutrophil, mononuclear cell, and *H. pylori* scores improved in all groups, but no significant differences in these scores were observed among the groups.

Associations between changes in the updated Sydney system scores and duration of PPI and H2RA use are shown in Table 3 and Table 4. Long-term PPI users showed significantly reduced improvement of corpus atrophy score compared to short-term users (*p* = 0.016). No significant differences in all Sydney system factor scores were observed between long- and short-term H2RA drug use.

### 2.3. Association between Caudal Related Homeobox Gene (CDX)1 and CDX2 Expression and Changes in Metaplasia and Atrophy Scores

We selected 63 patients whose metaplasia scores markedly changed after eradication (36 patients with improvement and 27 patients with exacerbation) and investigated changes in Caudal related homeobox gene (CDX)1 and CDX2 expression levels between the initial and final biopsies based on immunohistochemistry, and incomplete metaplasia rate in the same specimens upon hematoxylin and eosin staining.

CDX1 was expressed in 16 of 63 patients (25.4%). CDX1 expression level significantly decreased in patients showing an improvement in the metaplasia score of the antrum (0.217 to 0.033, *p* < 0.001) and corpus (0.092 to 0.031, *p* = 0.019) (Table 5).

CDX2 was expressed in 25 of 63 patients (39.7%). CDX2 expression level significantly decreased in patients with an improvement in the metaplasia score of the antrum (0.195 to 0.01, *p* < 0.001) (Table 5).

We also evaluated the changes in incomplete metaplasia prevalence in these specimens. The changes were mostly similar to intestinal metaplasia score improvement/exacerbation, but no statistically significant differences were observed (Appendix A).

In addition, we evaluated CDX1 and CDX2 expressional changes in cases with significant atrophy score changes, but no association with these molecular expression rates was observed (Appendix A).

## 3. Discussion

We showed that *H*. *pylori* eradication improves histological atrophy and inflammation but does not accelerate the recovery of intestinal metaplasia in our 5.4-year observation period. In particular, PPI or H2RA use may inhibit the improvement of gastric atrophy compared with non-acid suppressant users. Therefore, gastric atrophy may be regulated by molecularly distinct mechanisms to that of intestinal metaplasia (i.e., aberrant expression of CDX proteins). In other words, improvement of gastric atrophy after *H. pylori* eradication may be independent of changes in intestinal metaplasia [15,16,17,18,19,20], which seems to be a more sustained, stable reprogramming event of the gastric glands. In our study, the expression of CDX1 was more closely associated with changes in the intestinal metaplasia score than that of CDX2. In mice, ectopic expression of CDX1 and CDX2 in the stomach induces intestinal metaplasia, but expression of either CDX genes alone does not induce expression of the other gene [21]. Therefore, CDX1 and CDX2 may be independently regulated. Our findings suggest that changes in CDX1 expression might precede changes in CDX2 expression.

The finding that eradication resulted in improvement of atrophy in the corpus is consistent with a previous meta-analysis of short-term observational studies [22] and a long-term observational study in Japan [23] but inconsistent with a randomized controlled trial in Korea [15] and a long-term observational study [24]. By contrast, the lack of improvement in intestinal metaplasia after eradication is inconsistent with some previous studies [15,22]. This may be explained by the different scoring systems used for evaluation of atrophy and metaplasia. In addition, baseline histological severities of atrophy and intestinal metaplasia were different. There are also differences in regions and countries, races and, presumably, *H. pylori* strains among these studies, all of which may affect outcomes.

Previously, Annibale et al. reported that *H. pylori* eradication improved gastric atrophy only in a subset of patients whose serum gastrin level became normal during the observation period, while the remaining patients showed sustained gastric atrophy with elevated serum gastrin [25]. A reasonable interpretation of these results would be that restoration of normal acid secretion in improved patients suppresses the elevation of serum gastrin in a negative feedback manner. Nonetheless, it is possible that an elevated level of serum gastrin, perhaps due to the intake of acid-suppressing drugs, delays the recovery of gastric atrophy. In mice, hypergastrinemia induces gastric atrophy, metaplasia and, eventually, cancer [26], and these changes are accelerated by the administration of PPIs [27]. Long-term PPI use also promotes gastric atrophy and cancer development in *Helicobacter*-infected Mongolian gerbils [28]. In humans, long-term use of PPIs has been reported to worsen corpus atrophic gastritis in the patients infected with *H. pylori* [29,30]. In any case, prior epidemiological retrospective cohort studies [11,12] and our current findings suggest that PPI use may prolong gastric atrophy after eradication, which may increase the risk for gastric cancer in post-eradicated patients [1].

Although *H. pylori* infection appears to be inversely associated with gastroesophageal reflux disease due to low acid secretion from the atrophic stomach, certain populations of *H. pylori*-infected patients continue to suffer from acid reflux symptoms. In our cohort, the gastrointestinal symptom rating scale (GSRS) scores of upper abdominal pain, heartburn, abdominal distention, and bloating were temporarily decreased (improved) after *H. pylori* eradication (Appendix A), while these scores eventually returned to their original levels after the mean 5-year observation period. Thus, PPIs or H2RAs are often prescribed for patients with chronic acid reflux symptoms after eradication; however, our data suggest that the use of H2RAs or PPIs may be more cautiously considered given the potential effects on gastric atrophy and subsequent cancer risk.

The strengths of this study are the long-term, comprehensive assessment of clinical, endoscopic, histological, and molecular findings. Our analysis is quantitative and has extensively investigated the long-term effects of PPI and H2RA use following *H*. *pylori* eradication. However, this study also had limitations. Firstly, although we conducted long-term follow-up, the analyses were not prospective. The number of biopsy specimens and the observation period were limited, and some patients were lost to follow-up; secondly, our immunohistochemical stain data were limited to certain patients, associated with potential selection bias; thirdly, our sample size might be too small to reliably evaluate further associations between duration of drug use and histological score changes; fourthly, NSAIDs were more commonly taken in PPI and H2RA users than in non-drug users, and it might act as a confounder and affect the outcome; finally, our cohort lacked consideration of confounding factors such as smoking status, alcohol intake, and diet.

In conclusion, gastric atrophy but not intestinal metaplasia may be improved in patients who underwent successful *H*. *pylori* eradication, especially in patients with non-acid-suppressing drug use. Long-term acid suppression therapy might be associated with sustained gastric atrophy following eradication.

## 4. Methods

### 4.1. Study Design and Setting

We collected consecutive data on patients who underwent *H*. *pylori* eradication therapy and gastric cancer surveillance at Tokyo University Hospital from 1996 to 2015 (Figure 1A, study flow chart). We performed retrospective cohort analyses of these data. This study was approved by the Institutional Review Board of the University of Tokyo (no. 2058-2, 17 November 2017) and was performed in accordance with an assurance filed with and approved by the Japanese Ministry of Health, Labour, and Welfare.

### 4.2. Participants

Patients who were positive for *H*. *pylori* infection and atrophic gastritis by endoscopy, and had undergone successful eradication therapy, were eligible for inclusion in this study. *H*. *pylori* infection was diagnosed by urea breath test, culture, rapid urease test, or serum antibody test. An exclusion criterion was a non-eradication assessment by urea breath test.

### 4.3. Eradication Treatment and Assessment

As the first-line treatment, amoxicillin (750 mg), clarithromycin (500 mg), and the PPI lansoprazole (30 mg) were administered twice daily for 7 days. As the second-line treatment, metronidazole (250 mg), amoxicillin (750 mg), and the PPI lansoprazole (30 mg) were given twice daily for 7 days. The *H*. *pylori* eradication status was determined based on the urea breath test.

After eradication, all patients underwent upper gastrointestinal endoscopy evaluation every 1 year up to 2017. Data were censored at the time of the last endoscopic examination or the last hospital visit for patients who were lost to follow-up or who withdrew from the study (Figure 1B, study time course).

### 4.4. Upper Gastrointestinal Endoscopy and Histological Examination

All upper gastrointestinal endoscopies were performed using an electronic video endoscope (Olympus Medical System, Tokyo, Japan).

An endoscopic biopsy was performed to evaluate intestinal metaplasia, atrophy, neutrophil and mononuclear cell infiltration, and *H*. *pylori* scores according to the updated Sydney system [31]. These parameters were scored as follows: normal, 0 points; mild, 1 point; moderate, 2 points; and marked, 3 points. Biopsy specimens were obtained from the antrum and the middle corpus of the greater curvature based on our previous works [1,12,32]. Incomplete metaplasia was also evaluated by hematoxylin and eosin staining. Histological findings were assessed and confirmed by two experienced pathologists without any disagreement.

### 4.5. Outcomes and Variables

The primary outcome was change in the intestinal metaplasia, atrophy, neutrophil and mononuclear cell infiltration, and *H*. *pylori* scores in patients with PPI, H_2_ receptor agonist (H2RA), and non-acid suppressant use. The secondary outcome was change in these scores in patients according to the duration of acid suppressing therapy. We also evaluated associations in changes of histology and CDX1 and CDX2 expression between the first and final biopsies in the cases whose intestinal metaplasia or atrophy score changed during study periods. For intestinal metaplasia and atrophy, improvement and exacerbation were defined as a decrease and increase, respectively, in the updated Sydney system score between the first and final biopsies.

We evaluated age, sex, medication use (PPIs, H2RAs, and non-steroidal anti-inflammatory drugs (NSAIDs)), and acid-reflux symptoms. Age was categorized into five groups. Use of PPIs, H2RAs, and NSAIDs was defined as regular oral administration for at least 30 days during the follow-up period. Based on an appropriate receiver operating characteristic curve model for predicting atrophy execration, duration of PPIs and H2RAs was categorized as follows: PPI short-term use, <90 days; PPI long-term use, ≥90 days; H2RA short-term use, <485 days; H2RA long-term use, ≥485 days. Use of NSAIDs included low-dose aspirin use. Acid-reflux symptoms were evaluated using the upper gastrointestinal related gastrointestinal symptom rating scale (GSRS): upper abdominal pain, heartburn, acid reflux, abdominal distention, hunger pain, rumbling, bloating, and burning, administered as a seven-point Likert scale self-reported questionnaire [33].

### 4.6. Immunohistochemical Stain Analyses of CDX1 and CDX2 Expression

Sections (3 mm thick) were deparaffinized, rehydrated in phosphate-buffered saline (PBS), placed in 10 mM citrate buffer (pH 6.0), and heated to 120 °C for 5 min to recover antigenicity. Sections were preincubated with blocking buffer (3% hydrogen peroxide) for 5 min at room temperature. The primary anti-CDX1 polyclonal antibody (1:100; BioGenex, Fremont, CA, USA) and anti-CDX2 monoclonal antibody (1:100; Abcam, Cambridge, UK) were diluted in PBS and incubated with the sections overnight at 4 °C. Sections were then washed in PBS and incubated with Histofine Simple Stain Max-Po (Multi) (Nichirei Biosciences Inc., Tokyo, Japan). After development with 3,3,9-diaminobenzidine tetrahydrochloride (DAB Substrate Kit, Nichirei Biosciences), sections were counterstained with hematoxylin and viewed under a light microscope. CDX1 and CDX2 expression were assessed as proportions of CDX1- and CDX2-expressing gastric glands by one researcher (RN) (Figure 2).

### 4.7. Statistical Analyses

We calculated the coefficients and 95% confidence intervals (CIs) of annual changes in the atrophy, intestinal metaplasia, mononuclear cell and neutrophil infiltration, and *H*. *pylori* scores in biopsies of the antrum and middle corpus of the greater curvature and compared these scores among PPI, H2Rs, and non-acid suppressant users based on a generalized linear mixed model. We also compared the coefficients and 95% CIs of the annual changes of atrophy, intestinal metaplasia, mononuclear cell and neutrophil infiltration, and *H*. *pylori* scores between short- and long-term users of PPIs and H2RAs using a generalized linear mixed model. In addition, we evaluated changes between the initial and final CDX1 and CDX2 expression levels using paired *t*-tests in each metaplasia and atrophy change subgroup. Furthermore, we evaluated changes between the initial and final incomplete metaplasia rates using chi-squared tests in each metaplasia subgroup. We evaluated changes in the GSRS before and 3 months after eradication and changes in the GSRS during all follow-up periods using paired *t*-tests. Statistical analyses were performed using SAS software v. 9.4 (SAS Institute, Cary, NC, USA). A *p*-value < 0.05 was considered to indicate statistical significance.

## Figures and Tables

**Figure 1 ijms-20-01913-f001:**
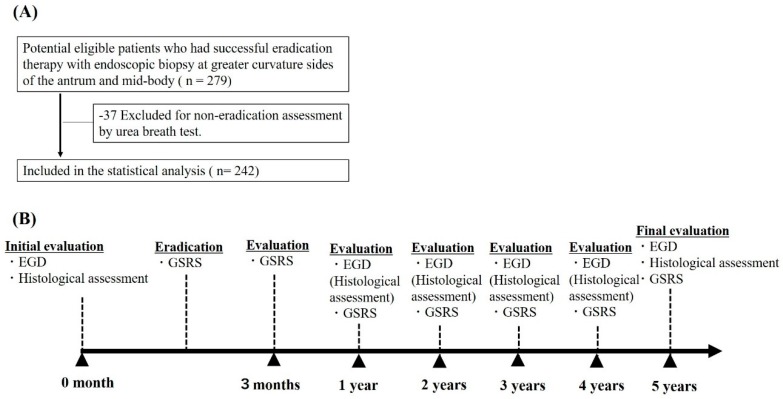
(**A**) Study flow chart; (**B**) Study time course. Abbreviation: EGD, Esophagogastroduodenoscopy; GSRS, gastrointestinal symptom rating scale.

**Figure 2 ijms-20-01913-f002:**
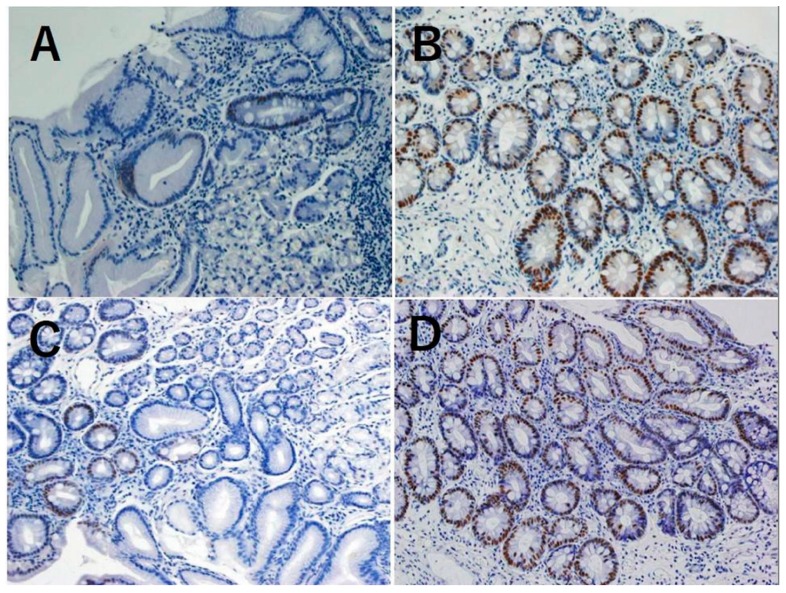
Immunohistochemical stain of Caudal related homeobox gene (CDX)1 and CDX2 expression in the biopsied gastric mucosa: low positive (<50%) for CDX1 100× (**A**); high positive (>50%) for CDX1 100× (**B**); low positive (<50%) for CDX2 100× (**C**); high positive (>50%) for CDX2 100× (**D**).

**Table 1 ijms-20-01913-t001:** Baseline patient characteristics by drug use (*n* = 242).

Characteristics	No. of Patients (%)			*p* Value
	PPI Use (*n* = 110)	H2RA Use (*n* = 33)	Non-Acid Suppressant Use (*n* = 99)	
Age category (years)				
<50	4 (3.63)	5 (15.15)	6 (6.06)	0.564
50–60	12 (10.91)	4 (12.13)	19 (19.19)	
60–70	37 (33.64)	11 (33.33)	37 (37.37)	
70–80	46 (41.82)	13 (39.39)	35 (35.35)	
≥80	11 (10.00)	0	2 (2.02)	
Sex				
Female	49 (44.55)	20 (60.61)	44 (44.44)	0.202
Male	61 (55.45)	13 (39.39)	55 (55.56)	
NSAID use	51 (46.36)	17 (51.52)	24 (24.24)	**<0.001**
Sydney System Factor Score				
Atrophy antrum	1.221 ± 0.778	1.160 ± 0.898	0.986 ± 0.778	0.254
corpus	0.755 ± 0.860	0.429 ± 0.898	0.613 ± 0.751	0.140
Metaplasia antrum	0.633 ± 0.889	0.515 ± 0.667	0.443 ± 0.667	0.411
corpus	0.284 ± 0.708	0.156 ± 0.448	0.122 ± 0.503	0.116
Mononuclear cell antrum	1.211 ± 0.639	1.303 ± 0.684	1.243 ± 0.648	0.713
corpus	1.211 ± 0.759	1.273 ± 0.674	1.102 ± 0.681	0.428
Neutrophil antrum	0.352 ± 0.646	0.455 ± 0.833	0.427 ± 0.778	0.925
corpus	0.444 ± 0.777	0.424 ± 0.708	0.433 ± 0.776	0.947
*H*. *pylori* antrum	0.389 ± 0.783	0.273 ± 0.626	0.333 ± 0.660	0.769
corpus	0.333 ± 0.641	0.424 ± 0.969	0.443 ± 0.841	0.826

Abbreviations: PPI, proton pump inhibitor; H2RA, histamine 2 receptor antagonist; NSAID, non-steroidal anti-inflammatory drug. Bold, statistical significance.

**Table 2 ijms-20-01913-t002:** Changes in the updated Sydney system score over time by drug use (*n* = 242).

Sydney System Factor	PPI Use (*n* = 110)	H2RA Use (*n* = 33)	Non-Acid Suppressant Use (*n* = 99)	*p* Value
	Score Coefficient (Point), Per Year (95% CI)			
Atrophy antrum	−0.004 (−0.028 to 0.020)	−0.011 (−0.060 to 0.039)	−0.036 (−0.059 to −0.012)	**0.042**
corpus	−0.023 (−0.045 to −0.0002)	−0.021 (−0.050 to 0.009)	−0.030 (−0.052 to −0.007)	**0.020**
Metaplasia antrum	0.014 (−0.009 to 0.036)	−0.031 (−0.069 to 0.007)	−0.007 (−0.030 to 0.015)	0.271
corpus	−0.003 (−0.020 to 0.013)	−0.007 (−0.033 to 0.019)	0.001 (−0.012 to 0.015)	0.077
Mononuclear cell antrum	−0.050 (−0.064 to −0.035)	−0.050 (−0.080 to −0.020)	−0.055 (−0.071 to −0.039)	0.815
corpus	−0.063 (−0.080 to −0.045)	−0.035 (−0.062 to −0.008)	−0.051 (−0.068 to −0.034)	0.204
Neutrophil antrum	−0.034 (−0.049 to −0.019)	−0.040 (−0.072 to −0.009)	−0.053 (−0.069 to −0.037)	0.961
corpus	−0.047 (−0.065 to −0.029)	−0.037 (−0.065 to −0.009)	−0.050 (−0.066 to −0.034)	0.350
*H*. *pylori* antrum	−0.038 (−0.055 to −0.020)	−0.027 (−0.052 to −0.001)	−0.043 (−0.057 to −0.028)	0.395
corpus	−0.040 (−0.056 to −0.023)	−0.042 (−0.082 to −0.003)	−0.060 (−0.078 to −0.041)	0.906

Abbreviations: PPI, proton pump inhibitor; H2RA, histamine 2 receptor antagonist. Bold, statistical significance.

**Table 3 ijms-20-01913-t003:** Subgroup analysis of changes in the updated Sydney system score over time in relation to duration of PPI use.

Sydney System Factor	PPI Long-term Use (*n* = 87)	PPI Short-term Use (*n* = 23)	*p* Value
	Score Coefficient (Point), Per Year (95% CI)		
Atrophy antrum	−0.002 (−0.029 to 0.026)	−0.016 (−0.075 to 0.043)	0.189
corpus	−0.021 (−0.045 to 0.003)	−0.027 (−0.095 to 0.041)	**0.016**
Metaplasia antrum	0.014 (−0.011 to 0.039)	0.006 (−0.048 to 0.060)	0.639
corpus	−0.0001 (−0.017 to 0.017)	−0.040 (−0.092 to 0.012)	0.177
Mononuclear cell antrum	−0.052 (−0.067 to −0.037)	−0.036 (−0.082 to 0.009)	0.957
corpus	−0.064 (−0.084 to −0.045)	−0.051 (−0.088 to −0.013)	0.656
Neutrophil antrum	−0.028 (−0.043 to −0.012)	−0.060 (−0.102 to −0.018)	0.682
corpus	−0.042 (−0.063 to −0.022)	−0.070 (−0.110 to −0.030)	0.777
*H*. *pylori* antrum	−0.032 (−0.049 to −0.014)	−0.067 (−0.128 to −0.006)	0.107
corpus	−0.038 (−0.057 to −0.019)	−0.044 (−0.078 to −0.010)	0.631

Abbreviations: PPIs, proton pump inhibitors. Bold, statistical significance.

**Table 4 ijms-20-01913-t004:** Subgroup analysis of changes in the updated Sydney system score over time in relation to duration of H2RA use.

Sydney System Factor	H2RA Long-term Use (*n* = 16)	H2RA Short-term Use (*n* = 17)	*p* Value
	Score Coefficient (Point), Per Year (95% CI)		
Atrophy antrum	−0.037 (−0.117 to 0.044)	0.045 (−0.056 to 0.147)	0.382
corpus	−0.034 (−0.067 to −0.001)	0.005 (−0.056 to 0.066)	0.108
Metaplasia antrum	−0.053 (−0.100 to −0.0057)	−0.022 (−0.075 to 0.031)	0.296
corpus	−0.021 (−0.066 to 0.024)	−0.008 (−0.035 to 0.019)	0.350
Mononuclear cell antrum	−0.039 (−0.085 to 0.007)	−0.078 (−0.135 to −0.022)	0.567
corpus	−0.026 (−0.070 to 0.017)	−0.051 (−0.102 to −0.001)	0.988
Neutrophil antrum	−0.034 (−0.076 to 0.009)	−0.066 (−0.129 to −0.002)	0.681
corpus	−0.026 (−0.061 to 0.009)	−0.063 (−0.124 to −0.003)	0.692
*H*. *pylori* antrum	−0.023 (−0.057 to 0.010)	−0.065 (−0.144 to 0.015)	0.443
corpus	−0.035 (−0.084 to 0.014)	−0.038 (−0.091 to 0.015)	0.633

Abbreviations: H2RA, histamine 2 receptor antagonist.

**Table 5 ijms-20-01913-t005:** Association between changes in the metaplasia score and CDX1 and CDX2 expression (N = 63).

**Metaplasia Change**	**CDX1 Expression Rates, Mean ± SD**		***p* Value**
	**Initial assessment**	**Final assessment**	
Improvement			
Antrum (N = 25)	0.217 ± 0.231	0.033 ± 0.091	**<0.001**
Corpus (N = 11)	0.092 ± 0.103	0.008 ± 0.022	**0.019**
Exacerbation			
Antrum (N = 21)	0.059 ± 0.132	0.100 ± 0.169	0.128
Corpus (N = 6)	0.002 ± 0.005	0.093 ± 0.105	0.080
**Metaplasia Change**	**CDX2 Expression Rate, Mean ± SD**		***p* Value**
	**Initial Assessment**	**Final Assessment**	
Improvement			
Antrum (N = 25)	0.195 ± 0.194	0.031 ± 0.059	**<0.001**
Corpus (N = 11)	0.128 ± 0.168	0.069 ± 0.211	0.401
Exacerbation			
Antrum (N = 21)	0.060 ± 0.100	0.114 ± 0.156	0.079
Corpus (N = 6)	0.003 ± 0.007	0.103 ± 0.155	0.159

Improvement and exacerbation were defined as a decrease and increase, respectively, in the updated Sydney system score between the first and final biopsies. Abbreviations: SD, standard deviation; CDX, Caudal related homeobox gene. Bold, statistical significance.

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
