# Peer review of "The Reduction in Gastric Atrophy after Helicobacter pylori Eradication Is Reduced by Treatment with Inhibitors of Gastric Acid Secretion"

_ijms, 2019, doi:10.3390/ijms20081913_

Reviewer 1 Report

This is a retrospective cohort study on impact of long-term acid suppression on histological atrophy progression after Helicobacter pylori eradication. The topic is interesting and clinically important; however, there are several points to be checked as follow.

 1. Title – Please consider changing the title to phrase, instead of writing in sentence.

 2. Abstract – Add numeric values in the results. For example, add SD in the mean follow-up period. Also add p-values at the end of sentences. Moreover, all of the sentences should be written in “past tense” in the results.

 3. Introduction, lines 65-66 - Please add recent studies on preventing effect on decreasing metachronous tumors after endoscopic resection for gastric neoplasm. Some examples are as follow.

(1) Shichijo S, et al. Histologic intestinal metaplasia and endoscopic atrophy are predictors of gastric cancer development after Helicobacter pylori eradication. Gastrointest Endosc. 2016 Oct;84(4):618-24.

(2) Murakami K, et al. Long-term monitoring of gastric atrophy and intestinal metaplasia after Helicobacter pylori eradication. Clin J Gastroenterol. 2012 Aug;5(4):247-50.

(3) Kodama M, et al. Helicobacter pylori eradication improves gastric atrophy and intestinal metaplasia in long-term observation. Digestion. 2012;85(2):126-30.

(4) Toyokawa T, et al. Eradication of Helicobacter pylori infection improved gastric mucosal atrophy and prevented progression of intestinal metaplasia, especially in the elderly population: a long-term prospective cohort study. J Gastroenterol Hepatol. 2010 Mar;25(3):544-7.

(5) Rocco A, et al. Gastric atrophy and intestinal metaplasia changes 8 years after Helicobacter pylori eradication. A blind, randomised study. Minerva Gastroenterol Dietol. 2002 Jun;48(2):175-8.

4. Methods – Main flaw of this study might be wrong inclusion and exclusion criteria. Only successfully eradicated subjects should be included; and therefore, 12 subjects who were not confirmed after Hp eradication should be excluded. To strengthen the study, please analyze only 273 subjects NOT 285 subjects.

5. Methods, Participants - It is written that the subjects were included according to the presence of Hp and atrophy in the endoscopic biopsied specimens. To make things clear, total number of the subjects who underwent endoscopic biopsy at two different sites (GC sides of the antrum and mid-body) between 1996 and 2015 should be shown in the study flow (Figure 1). This will elucidate the proportion of subjects who had Hp infection and gastric atrophy at the time of inclusion.

6. Methods, lines 104-105 – As recommended by the European guideline, CLO test should not be used for the confirmation of successful Hp eradication. So please consider including only the eradicated subjects confirmed by UBT. Rapid urease test is inaccurate when evaluating the status of Hp after eradication.

7. Methods, lines 118-119 – Was this study approved as a prospective study by the IRB? If not, how can the endoscopists take biopsies at the same sites in all the subjects during the study period?

8. Methods, lines 119-120 – The sentence is too vague. How many pathologists? What consensus? What kind of disagreement?

9. Results – Please classifying the subjects into three groups (subjects with PPI vs. with H2 blocker vs. controls), and show their baseline characteristics in Table 1. Thereafter, the changes in the degree of atrophy and intestinal metaplasia should be compared in each group using the signed-rank test. If the findings are statistically significant, details of PPI and H2RA (dose, duration, type, etc.) should be described in detail.

10. Table 1 - Instead of summarizing in age groups or endoscopic atrophy (C1-O3), the baseline characteristics of the subjects (PPI group vs. H2RA group vs. controls) should be shown. Please verify whether there was a difference in the degree of atrophy or IM (updated Sydney system scores in the antrum and body) between the three groups at the initiation of this study.

11. Results, line 191 - Only 73 underwent IHC, so please do not focus on cdx1 and cdx2 IHC findings. There might be a selection bias. Describe how the authors selected these subjects. Also, please mention about the type of IM (complete vs. incomplete).

12. Discussion, line 248 – Because there is no data on the duration of symptoms in this study, “temporarily” should be deleted.

13. Discussion, lines 261-263 – Current findings are insufficient to draw the conclusions. Please revise the conclusions after analyzing the different changes on the degree of atrophy and intestinal metaplasia between the three groups.

14. References 12 and 17 are inappropriate. Please consider revising them to other papers.

15. Table 4 – The only difference between cdx1 and cdx2 was IM improvement in the corpus. Please explain in the Discussion why the difference was found only in cdx1.

16. Figure 1 – Figure legends should be written in sentences.

17. Supplementary Table 3 – Please show the correlations between symptom scores and pathologic findings first.

18. Authorship statements, lines 266-270 – There is no description on study performance. For example, who performed endoscopic biopsies?

19. Lastly, English language editing is required.

Author Response

#Reviewer1

This is a retrospective cohort study on impact of long-term acid suppression on histological atrophy progression after Helicobacter pylori eradication. The topic is interesting and clinically important; however, there are several points to be checked as follow.

1. Title – Please consider changing the title to phrase, instead of writing in sentence.

Response: We have changed the title to “Inhibition of the improvement of histological gastric atrophy by long-term acid suppression in patients after Helicobacter pylori eradication”.

2. Abstract – Add numeric values in the results. For example, add SD in the mean follow-up period. Also add p-values at the end of sentences. Moreover, all of the sentences should be written in “past tense” in the results.

Response: Thanks for detailed comments. We have added SDs to the mean follow-up periods, and p-values. In addition, we have changed to past tense in the Abstract.

“The mean follow-up period and number of biopsies were 5.48 ± 4.69 years and 2.62 ± 1.67 times, respectively. Improvement in the atrophy scores of both in antrum (p = 0.042) and corpus (p = 0.020) were significantly superior in patients with no-acid suppressant use compared with those of PPIs and H2RAs use. Metaplasia scores in both antrum and corpus did not improve in all groups, and no significant differences were observed among groups in antrum (p = 0.271) and corpus (p = 0.077).”

3. Introduction, lines 65-66 - Please add recent studies on preventing effect on decreasing metachronous tumors after endoscopic resection for gastric neoplasm. Some examples are as follow.

(1) Shichijo S, et al. Histologic intestinal metaplasia and endoscopic atrophy are predictors of gastric cancer development after Helicobacter pylori eradication. Gastrointest Endosc. 2016 Oct;84(4):618-24.

(2) Murakami K, et al. Long-term monitoring of gastric atrophy and intestinal metaplasia after Helicobacter pylori eradication. Clin J Gastroenterol. 2012 Aug;5(4):247-50.

(3) Kodama M, et al. Helicobacter pylori eradication improves gastric atrophy and intestinal metaplasia in long-term observation. Digestion. 2012;85(2):126-30.

(4) Toyokawa T, et al. Eradication of Helicobacter pylori infection improved gastric mucosal atrophy and prevented progression of intestinal metaplasia, especially in the elderly population: a long-term prospective cohort study. J Gastroenterol Hepatol. 2010 Mar;25(3):544-7.

(5) Rocco A, et al. Gastric atrophy and intestinal metaplasia changes 8 years after Helicobacter pylori eradication. A blind, randomised study. Minerva Gastroenterol Dietol. 2002 Jun;48(2):175-8.

Response: We have added these references #7-#11 (Introduction, lines 65-66).

4. Methods – Main flaw of this study might be wrong inclusion and exclusion criteria. Only successfully eradicated subjects should be included; and therefore, 12 subjects who were not confirmed after Hp eradication should be excluded. To strengthen the study, please analyze only 273 subjects NOT 285 subjects.

Response: Thank you for this important comment. We agree and have re-analyzed the data of the 242 patients with successful eradication confirmed by urea breath test and revised the manuscript (please confirm item 6 response).

5. Methods, Participants - It is written that the subjects were included according to the presence of Hp and atrophy in the endoscopic biopsied specimens. To make things clear, total number of the subjects who underwent endoscopic biopsy at two different sites (GC sides of the antrum and mid-body) between 1996 and 2015 should be shown in the study flow (Figure 1). This will elucidate the proportion of subjects who had Hp infection and gastric atrophy at the time of inclusion.

Response: A total of 285 subjects underwent endoscopic biopsy at two different sites (the GC sides of the antrum and mid-body) between 1996 and 2015. We used data from eligible patients who had successful eradication therapy with assessment by urea breath test in the analysis. To make these clear, we have changed Figure 1.

6. Methods, lines 104-105 – As recommended by the European guideline, CLO test should not be used for the confirmation of successful Hp eradication. So please consider including only the eradicated subjects confirmed by UBT. Rapid urease test is inaccurate when evaluating the status of Hp after eradication.

Response: As you suggest, we included only patients in whom eradication was confirmed by UBT (n = 242). We have changed the study flow in Figure 1 accordingly, and have re-analyzed the data.

7. Methods, lines 118-119 – Was this study approved as a prospective study by the IRB? If not, how can the endoscopists take biopsies at the same sites in all the subjects during the study period?

Response: This is a retrospective study approved by our IRB. In our country, H. pylori-infected patients and eradicated patients are generally recommended to undergo annual endoscopic gastric cancer surveillance. In our facility, all endoscopists were carefully trained in appropriate endoscopic photography, and in the collection of appropriate biopsy samples. We ensured that the gastric cancer surveillance was accurate. Technically, specialized endoscopists do not find it difficult to re-biopsy previous sites; they recognize the mucosal scars. We earlier reported that the extent of histological atrophy, and of intestinal metaplasia, strongly predicted the risk of gastric cancer in gastritis patients, and we thus routinely perform background biopsies of the body and antrum to evaluate the subsequent cancer risk and optimize surveillance. However, we do not do this if a patient have any contraindication for biopsy [Gastrointestinal Endoscopy 2016;84:618-624; Journal of Gastroenterology and Hepatology 2015;30:1260–1264; Gut 2018;67:1908-1910.]. We obtain a written form of informed consent from all patients at every endoscopy examination.

8. Methods, lines 119-120 – The sentence is too vague. How many pathologists? What consensus? What kind of disagreement?

Response: We apologize for our imprecision. Two pathologists (one junior and one senior) evaluated all gastric specimens achieved consensus using the updated Sydney system. In several specimens, it was difficult to evaluate atrophy because of small specimen size and a lack of any muscularis mucosa. In such cases, the two pathologists discussed their evaluations and reached to a conclusion. We have changed text as follows:

“Histological findings were assessed and confirmed by two experienced pathologists without any disagreement.” (Methods, lines 119-120)

9. Results – Please classifying the subjects into three groups (subjects with PPI vs. with H2 blocker vs. controls), and show their baseline characteristics in Table 1. Thereafter, the changes in the degree of atrophy and intestinal metaplasia should be compared in each group using the signed-rank test. If the findings are statistically significant, details of PPI and H2RA (dose, duration, type, etc.) should be described in detail.

Response: We classified all patients into users of PPIs, H2RAs users, and no-acid suppressant use. We have re-analyzed the data and changed the Tables. Please see item 10 response.

10. Table 1 - Instead of summarizing in age groups or endoscopic atrophy (C1-O3), the baseline characteristics of the subjects (PPI group vs. H2RA group vs. controls) should be shown. Please verify whether there was a difference in the degree of atrophy or IM (updated Sydney system scores in the antrum and body) between the three groups at the initiation of this study.

Response: As mentioned above, we have changed Table 1 as follows:

Table 1. Baseline patient characteristics by drug use (N = 242)

No. of patients (%)

Characteristics

PPIs use (N=110)

H2RAs use (N = 33)

No-acid suppressant use (N = 99)

P value

Age category   (years)

< 50

4   (3.63)

5   (15.15)

6   (6.06)

0.564

50–60

12   (10.91)

4   (12.13)

19   (19.19)

60–70

37   (33.64)

11   (33.33)

37   (37.37)

70–80

46   (41.82)

13   (39.39)

35   (35.35)

≥ 80

11   (10.00)

0

2   (2.02)

Sex

Female

49   (44.55)

20   (60.61)

44   (44.44)

0.202

Male

61   (55.45)

13   (39.39)

55   (55.56)

NSAID use

51   (46.36)

17   (51.52)

24   (24.24)

<0.001< strong="">

Sydney system factor score

Atrophy   antrum

1.221   ± 0.778

1.160   ± 0.898

0.986   ± 0.778

0.254

corpus

0.755   ± 0.860

0.429   ± 0.898

0.613   ± 0.751

0.140

Metaplasia   antrum

0.633   ± 0.889

0.515   ± 0.667

0.443   ± 0.667

0.411

corpus

0.284   ± 0.708

0.156   ± 0.448

0.122   ± 0.503

0.116

Mononuclear   cells antrum

1.211   ± 0.639

1.303   ± 0.684

1.243   ± 0.648

0.713

corpus

1.211   ± 0.759

1.273   ± 0.674

1.102   ± 0.681

0.428

Neutrophil   antrum

0.352   ± 0.646

0.455   ± 0.833

0.427   ± 0.778

0.925

corpus

0.444   ± 0.777

0.424   ± 0.708

0.433   ± 0.776

0.947

H. pylori antrum

0.389   ± 0.783

0.273   ± 0.626

0.333   ± 0.660

0.769

corpus

0.333   ± 0.641

0.424   ± 0.969

0.443   ± 0.841

0.826

Abbreviations: PPIs, proton pump inhibitors; H2RAs, histamine 2 receptor antagonists; NSAID, non-steroidal anti-inflammatory drugs.

Bold, statistical significance.

Table 2. Changes in the updated Sydney system score over time by drug use (N = 242)

PPIs   use (N=110)

H2RAs   use (N = 33)

No-acid   suppressant use (N = 99)

P   value

Sydney   System factor

Score   coefficient (point), per year (95%CI)

Atrophy antrum

-0.004 (-0.028 to 0.020)

-0.011 (-0.060 to 0.039)

-0.036 (-0.059 to -0.012)

0.042

corpus

-0.023 (-0.045 to -0.0002)

-0.021 (-0.050 to 0.009)

-0.030 (-0.052 to -0.007)

0.020

Metaplasia antrum

0.014 (-0.009 to 0.036)

-0.031 (-0.069 to 0.007)

-0.007 (-0.030 to 0.015)

0.271

corpus

-0.003 (-0.020 to 0.013)

-0.007 (-0.033 to 0.019)

0.001 (-0.012 to 0.015)

0.077

Mononuclear cells antrum

-0.050 (-0.064 to -0.035)

-0.050 (-0.080 to -0.020)

-0.055 (-0.071 to -0.039)

0.815

corpus

-0.063 (-0.080 to -0.045)

-0.035 (-0.062 to -0.008)

-0.051 (-0.068 to -0.034)

0.204

Neutrophil antrum

-0.034 (-0.049 to -0.019)

-0.040 (-0.072 to -0.009)

-0.053 (-0.069 to -0.037)

0.961

corpus

-0.047 (-0.065 to -0.029)

-0.037 (-0.065 to -0.009)

-0.050 (-0.066 to -0.034)

0.350

H.   pylori antrum

-0.038 (-0.055 to -0.020)

-0.027 (-0.052 to -0.001)

-0.043 (-0.057 to -0.028)

0.395

corpus

-0.040 (-0.056 to -0.023)

-0.042 (-0.082 to -0.003)

-0.060 (-0.078 to -0.041)

0.906

Abbreviations: PPIs, proton pump inhibitors; H2RAs, histamine 2 receptor antagonists.

Bold, statistical significance.

Table 3. Subgroup analysis of duration of PPIs use of changes in the updated Sydney system score over time.

PPIs long term use (N=43)

PPIs short term use (N =67)

P value

Sydney system factor

Score coefficient (point), per year (95%CI)

Atrophy   antrum

-0.009   (-0.046 to 0.027)

-0.0008   (-0.036 to 0.034)

0.213

corpus

-0.028   (-0.060 to 0.005)

-0.019   (-0.052 to 0.015)

0.065

Metaplasia   antrum

0.011   (-0.020 to 0.042)

0.010   (-0.024 to 0.044)

0.960

corpus

-0.007   (-0.030 to 0.017)

-0.010   (-0.033 to 0.013)

0.691

Mononuclear   cells antrum

-0.055   (-0.073 to -0.037)

-0.039   (-0.063 to -0.015)

0.522

corpus

-0.071   (-0.098 to -0.044)

-0.057   (-0.082 to -0.031)

0.301

Neutrophil   antrum

-0.022   (-0.037 to -0.007)

-0.044   (-0.071 to -0.018)

0.742

corpus

-0.032   (-0.057 to -0.007)

-0.070   (-0.099 to -0.042)

0.607

H. pylori antrum

-0.019   (-0.033 to -0.005)

-0.057   (-0.089 to -0.024)

0.016

corpus

-0.027   (-0.045 to -0.008)

-0.056   (-0.084 to -0.027)

0.557

Abbreviations: PPIs, proton pump inhibitors.

Bold, statistical significance.

Table 4. Subgroup analysis of duration of H2RAs use of changes in the updated Sydney system score over time.

H2RAs long term use (N=12)

H2RAs short term use (N = 21)

P value

Sydney system factor

Score coefficient (point), per year (95%CI)

Atrophy   antrum

-0.058   (-0.148 to 0.033)

0.032   (-0.058 to 0.122)

0.088

corpus

-0.035   (-0.072 to 0.003)

-0.007   (-0.062 to 0.048)

0.056

Metaplasia   antrum

-0.045   (-0.100 to 0.009)

-0.029   (-0.088 to 0.030)

0.452

corpus

Not   applicable

-0.025   (-0.066 to 0.016)

0.734

Mononuclear   cells antrum

-0.036   (-0.090 to 0.017)

-0.091   (-0.143 to -0.039)

0.203

corpus

-0.021   (-0.064 to 0.021)

-0.071   (-0.120 to -0.023)

0.773

Neutrophil   antrum

-0.045   (-0.099 to 0.009)

-0.061   (-0.114 to -0.007)

0.307

corpus

-0.035   (-0.079 to 0.009)

-0.059   (-0.110 to -0.008)

0.316

H. pylori antrum

-0.026   (-0.068 to 0.016)

-0.038   (-0.084 to 0.007)

0.307

corpus

-0.047   (-0.110 to 0.016)

-0.060   (-0.127 to 0.007)

0.310

Abbreviations: H2RAs, histamine 2 receptor antagonists.

11. Results, line 191 - Only 73 underwent IHC, so please do not focus on cdx1 and cdx2 IHC findings. There might be a selection bias. Describe how the authors selected these subjects. Also, please mention about the type of IM (complete vs. incomplete).

Response: We agree that there might be a selection bias; we now mention this limitation in the Discussion. Ideally, IHC should be performed on all patients who undergo biopsies, but this would require us to perform over 1,000 IHCs, most of which would likely be “negative controls” lacking any changes in atrophy/metaplasia scores. To save effort and cost, we selected only patients whose metaplasia scores changed markedly among 2 different time points after eradication to the IHC analysis. We hope that the reviewer understands the situation.

We do not focus particularly on IHC data; we rather include these in our broader analyses. Although our IHC data are limited, we believe that the results should be of interest to particularly pathologists and biologists, because it remains unclear how CDX expression changes over time after H. pylori eradication.

We have added data on the type of intestinal metaplasia (complete or incomplete) to Supplementary Table 3.

Supplementary Table 3. Association between changes in the metaplasia score and incomplete metaplasia (N = 63).

Metaplasia change

Case with incomplete metaplasia

P-value

Initial assessment

Final assessment

Improvement

Antrum   (N = 25)

8   (32.00%)

3 (12.00   %)

0.215

Corpus   (N = 11)

5   (45.45%)

4 (36.36   %)

1.000

Exacerbation

Antrum   (N = 21)

4   (19.05%)

11   (52.38%)

0.090

Corpus   (N = 6)

3   (50.00%)

1   (16.67%)

1.000

12. Discussion, line 248 – Because there is no data on the duration of symptoms in this study, “temporarily” should be deleted.

Response. We apologize that we did not clearly present the data on the duration of symptoms. We showed data on the duration of symptoms in Supplementary Tables 4 and 5. We evaluated GSRS symptom scores at 3 months after eradication (Supplementary Table 4). Next, we evaluated GSRS symptom scores at all observational timepoints after eradication (thus during all follow-up periods) (Supplementary Table 5). Although pain/discomfort (GSRS1), heartburn (GSRS2), hunger pains (GSRS4), and bloating (GSRS7) improved during the 3 months after eradication, such improvements were not observed during all follow-up periods.

13. Discussion, lines 261-263 – Current findings are insufficient to draw the conclusions. Please revise the conclusions after analyzing the different changes on the degree of atrophy and intestinal metaplasia between the three groups.

Response: We have re-analyzed the data and changed conclusions as follows:

In conclusion, Gastric atrophy but not intestinal metaplasia may be improved in patients who had successful H. pylori eradication. Long-term acid suppression may be associated with sustained gastric atrophy following eradication.”

14. References 12 and 17 are inappropriate. Please consider revising them to other papers.

Response: Thank you for the comment; we have deleted these references.

15. Table 4 – The only difference between cdx1 and cdx2 was IM improvement in the corpus. Please explain in the Discussion why the difference was found only in cdx1.

Response: We now add the following in the Discussion :

“In mice, ectopic expression of CDX1 and CDX2 in the stomach induces intestinal metaplasia, but expression of either CDX genes alone does not induce expression of the other gene [Gut 2004 Oct;53(10):1416-1423]. Therefore, CDX1 and CDX2 may be independently regulated. Our findings suggest that changes in CDX1 expression might precede changes in CDX2 expression.

16. Figure 1 – Figure legends should be written in sentences.

Response: We have added Figure 1 legends in sentences.

17. Supplementary Table 3 – Please show the correlations between symptom scores and pathologic findings first.

Response: Thank you for this important comment. However, we suggest that additional analyses of symptom scores are beyond the scope of the study and may confuse readers. Our primary focus is the time course of pathology after H. pylori eradication, and the potential association between the changes and the use of acid suppression drugs. In future, we plan to explore the correlation that the reviewer suggests.    

18. Authorship statements, lines 266-270 – There is no description on study performance. For example, who performed endoscopic biopsies?

Response: Thank you. We have added a description of study performance as follows:

“R. Niikura, Y. Hayakawa, Y Hirata, K Ogura, and A Yamada performed endoscopic biopsies. R. Niikura and Y. Hayakawa conducted all immunohistochemical analyses.”

19. Lastly, English language editing is required.

Response: Our manuscript underwent professional English language editing (please see http://www.textcheck.com/text/page/about); this enhanced our revision. 

Reviewer 2 Report

As yet the paper cannot be fully evaluated as Figure 1 is missing.  Whilst ensuring that the full paper is submitted next time, I would suggest the authors also consider the following:

Reference/details of other potential confounding variables such as inclusion of smoking status of patients, alcohol intake, other medications such as NSAIDs, diet etc should be made for the cohorts studied.

Provide details of the length of time that patients took PPI or H2RAs, and whether length of time of medication correlated with influence on gastric atrophy.  What are the guidelines for duration of prescribing and were they not followed?

Provide some immunohistochemical figures rather that just referring to them.  The technique itself is not without limitations when considering antigen presentation and associated quantitation of antibody binding.

Author Response

#Reviewer 2

As yet the paper cannot be fully evaluated as Figure 1 is missing. Whilst ensuring that the full paper is submitted next time, I would suggest the authors also consider the following:

Response: We apologize. We uploaded Figure 1 as an EPS file in the first submission; this may have created a visibility issue. We have uploaded Figure 1 in a more readable format.   

1. Reference/details of other potential confounding variables such as inclusion of smoking status of patients, alcohol intake, other medications such as NSAIDs, diet etc should be made for the cohorts studied.

Response: Unfortunately, we lacked detailed data on smoking status, alcohol intake, and diet; we now mention this limitation in the Discussion. However, we have medication data such as NSAID. We have added data on NSAID in Table 1 which was revised by drug use groups based on another reviewer comments.

Table 1. Baseline patient characteristics by drug use (N = 242)

No. of patients (%)

Characteristics

PPIs use (N=110)

H2RAs use (N = 33)

No-acid suppressant drug (N = 99)

P value

Age category   (years)

< 50

4   (3.63)

5   (15.15)

6   (6.06)

0.564

50–60

12   (10.91)

4   (12.13)

19   (19.19)

60–70

37   (33.64)

11   (33.33)

37   (37.37)

70–80

46   (41.82)

13   (39.39)

35   (35.35)

≥ 80

11   (10.00)

0

2   (2.02)

Sex

Female

49   (44.55)

20   (60.61)

44   (44.44)

0.202

Male

61   (55.45)

13   (39.39)

55   (55.56)

NSAID use

51   (46.36)

17   (51.52)

24   (24.24)

<0.001< strong="">

Sydney system factor score

Atrophy   antrum

1.221   ± 0.778

1.160   ± 0.898

0.986   ± 0.778

0.254

corpus

0.755   ± 0.860

0.429   ± 0.898

0.613   ± 0.751

0.140

Metaplasia   antrum

0.633   ± 0.889

0.515   ± 0.667

0.443   ± 0.667

0.411

corpus

0.284   ± 0.708

0.156   ± 0.448

0.122   ± 0.503

0.116

Mononuclear   cells antrum

1.211   ± 0.639

1.303   ± 0.684

1.243   ± 0.648

0.713

corpus

1.211   ± 0.759

1.273   ± 0.674

1.102   ± 0.681

0.428

Neutrophil   antrum

0.352   ± 0.646

0.455   ± 0.833

0.427   ± 0.778

0.925

corpus

0.444   ± 0.777

0.424   ± 0.708

0.433   ± 0.776

0.947

H. pylori antrum

0.389   ± 0.783

0.273   ± 0.626

0.333   ± 0.660

0.769

corpus

0.333   ± 0.641

0.424   ± 0.969

0.443   ± 0.841

0.826

Abbreviations: PPIs, proton pump inhibitors; H2RAs, histamine 2 receptor antagonists; NSAID, non-steroidal anti-inflammatory drugs.  

Bold, statistical significance.

2-1. Provide details of the length of time that patients took PPI or H2RAs, and whether length of time of medication correlated with influence on gastric atrophy.

Response: As suggested, we have added data of length of time of PPIs and H2RAs, and performed additional analyses of association between medication duration and gastric atrophy (Table 3 and 4). In addition, we reanalyzed histological annual changes by drug use groups (PPIs, H2RAs, and no-acid suppressant use) based on another reviewer comments (Table 2).

Table 2. Changes in the updated Sydney system score over time by drug use (N = 242)

PPIs use (N=110)

H2RAs use (N = 33)

No-acid suppressant (N = 99)

P value

Sydney system factor

Score coefficient (point), per year (95%CI)

Atrophy   antrum

-0.004   (-0.028 to 0.020)

-0.011   (-0.060 to 0.039)

-0.036   (-0.059 to -0.012)

0.042

corpus

-0.023   (-0.045 to -0.0002)

-0.021   (-0.050 to 0.009)

-0.030   (-0.052 to -0.007)

0.020

Metaplasia   antrum

0.014   (-0.009 to 0.036)

-0.031   (-0.069 to 0.007)

-0.007   (-0.030 to 0.015)

0.271

corpus

-0.003   (-0.020 to 0.013)

-0.007   (-0.033 to 0.019)

0.001   (-0.012 to 0.015)

0.077

Mononuclear   cells antrum

-0.050   (-0.064 to -0.035)

-0.050   (-0.080 to -0.020)

-0.055   (-0.071 to -0.039)

0.815

corpus

-0.063   (-0.080 to -0.045)

-0.035   (-0.062 to -0.008)

-0.051   (-0.068 to -0.034)

0.204

Neutrophil   antrum

-0.034   (-0.049 to -0.019)

-0.040   (-0.072 to -0.009)

-0.053   (-0.069 to -0.037)

0.961

corpus

-0.047   (-0.065 to -0.029)

-0.037   (-0.065 to -0.009)

-0.050   (-0.066 to -0.034)

0.350

H. pylori antrum

-0.038   (-0.055 to -0.020)

-0.027   (-0.052 to -0.001)

-0.043   (-0.057 to -0.028)

0.395

corpus

-0.040   (-0.056 to -0.023)

-0.042   (-0.082 to -0.003)

-0.060   (-0.078 to -0.041)

0.906

Abbreviations: PPIs, proton pump inhibitors; H2RAs, histamine 2 receptor antagonists.

Bold, statistical significance.

Table 3. Subgroup analysis of duration of PPIs use of changes in the Updated Sydney System score over time.

PPIs long term use (N=43)

PPIs short term use (N =67)

P value

Sydney system factor

Score coefficient (point), per year (95%CI)

Atrophy   antrum

-0.009   (-0.046 to 0.027)

-0.0008   (-0.036 to 0.034)

0.213

corpus

-0.028   (-0.060 to 0.005)

-0.019   (-0.052 to 0.015)

0.065

Metaplasia   antrum

0.011   (-0.020 to 0.042)

0.010   (-0.024 to 0.044)

0.960

corpus

-0.007   (-0.030 to 0.017)

-0.010   (-0.033 to 0.013)

0.691

Mononuclear   cells antrum

-0.055   (-0.073 to -0.037)

-0.039   (-0.063 to -0.015)

0.522

corpus

-0.071   (-0.098 to -0.044)

-0.057   (-0.082 to -0.031)

0.301

Neutrophil   antrum

-0.022   (-0.037 to -0.007)

-0.044   (-0.071 to -0.018)

0.742

corpus

-0.032   (-0.057 to -0.007)

-0.070   (-0.099 to -0.042)

0.607

H. pylori antrum

-0.019   (-0.033 to -0.005)

-0.057   (-0.089 to -0.024)

0.016

corpus

-0.027   (-0.045 to -0.008)

-0.056   (-0.084 to -0.027)

0.557

Abbreviations: PPIs, proton pump inhibitors.

Bold, statistical significance.

Table 4. Subgroup analysis of duration of H2RAs use of changes in the updated Sydney system score over time.

H2RAs long term use (N=12)

H2RAs short term use (N = 21)

P value

Sydney System factor

Score coefficient (point), per year (95%CI)

Atrophy   antrum

-0.058   (-0.148 to 0.033)

0.032   (-0.058 to 0.122)

0.088

corpus

-0.035   (-0.072 to 0.003)

-0.007   (-0.062 to 0.048)

0.056

Metaplasia   antrum

-0.045   (-0.100 to 0.009)

-0.029   (-0.088 to 0.030)

0.452

corpus

Not   applicable

-0.025   (-0.066 to 0.016)

0.734

Mononuclear   cells antrum

-0.036   (-0.090 to 0.017)

-0.091   (-0.143 to -0.039)

0.203

corpus

-0.021   (-0.064 to 0.021)

-0.071   (-0.120 to -0.023)

0.773

Neutrophil   antrum

-0.045   (-0.099 to 0.009)

-0.061   (-0.114 to -0.007)

0.307

corpus

-0.035   (-0.079 to 0.009)

-0.059   (-0.110 to -0.008)

0.316

H. pylori antrum

-0.026   (-0.068 to 0.016)

-0.038   (-0.084 to 0.007)

0.307

corpus

-0.047   (-0.110 to 0.016)

-0.060   (-0.127 to 0.007)

0.310

Abbreviations: H2RAs, histamine 2 receptor antagonists.

2-2. What are the guidelines for duration of prescribing and were they not followed?

Response: Prescription was performed by attending physicians at our hospital based on patient symptoms; no particular guidelines were followed.

3. Provide some immunohistochemical figures rather that just referring to them. The technique itself is not without limitations when considering antigen presentation and associated quantitation of antibody binding.

Response: We have added Figure 2, which includes immunohistochemical images of CDX1 and CDX2.  Immunohistochemical of CDX1 and 2 expression in the biopsied gastric mucosa: low positive (< 50%) for CDX1 (A); high positive (> 50%) for CDX1 (B); low positive (< 50%) for CDX2 (C); high positive (> 50%) for CDX2

Round  2

Reviewer 1 Report

Thank you for submitting after revision. 

It has improved a lot; however, there are several issues to be checked further. 

With regard to the subgroup analysis of "short-term use" and "long-term use", the cutoff duration should be re-evaluated using the ROC curve analysis. Using the mean value seems inappropriate, because the duration showed significant correlation with the study outcomes. Therefore, the cutoff duration linked to atrophy/ IM improvement should be calculated using the ROC curves. Please revise Tables 3 and 4 as well.

Despite the hard work, the conclusions make this study less valuable. Because changes in IM were also noticed in considerable number of subjects, it should not be discriminated from atrophy. There are lot of recent studies showing that IM can be also improved after eradication. Therefore, the authors should better focus more on the duration of acid suppressants that diminished the effect of Hp eradication with regard to the improvement of atrophy and/or IM. ROC curve analysis might provide a solution for this.

There are still several typos including line 59 Helicobacter eradication -> H. pylori eradication, line 156 quintiles, line 577 "stain" missing after immunohistochemical, etc.

Figure 1 (A) - Please consider removing the last box in the bottom. The readers may think that only 63 subjects completed this study among the 242 included subjects. Large number of drop-out rate might weaken this study.

Please minimize the numbers of keywords. 

Author Response

#Reviewer1

Thank you for submitting after revision. It has improved a lot; however, there are several issues to be checked further.

1. With regard to the subgroup analysis of "short-term use" and "long-term use", the cutoff duration should be re-evaluated using the ROC curve analysis. Using the mean value seems inappropriate, because the duration showed significant correlation with the study outcomes. Therefore, the cutoff duration linked to atrophy/ IM improvement should be calculated using the ROC curves. Please revise Tables 3 and 4 as well.

Response: Thanks for detailed comments. We have calculated the appropriate cutoff duration of PPIs and H2RAs linked to atrophy execration using the ROC curve. We used the new cutoff duration of PPIs (PPIs short term use,<90 days; PPIs long term use, ≥ 90 days), H2RAs (H2RAs short term use, < 485 days, H2RAs long term use, ≥ 485 days) based on the highest youden index. We have revised Table 3 and 4 as follows, and revised the text in Result section.

Table 3. Subgroup analysis of duration of PPIs use of changes in the updated Sydney system score over time.

PPIs long term use (N=87)

PPIs short term use (N =23)

P value

Sydney system factor

Score coefficient (point), per   year (95%CI)

Atrophy antrum

-0.002 (-0.029 to 0.026)

-0.016 (-0.075 to 0.043)

0.189

corpus

-0.021 (-0.045 to 0.003)

-0.027 (-0.095 to 0.041)

0.016

Metaplasia antrum

0.014 (-0.011 to 0.039)

0.006 (-0.048 to 0.060)

0.639

corpus

-0.0001 (-0.017 to 0.017)

-0.040 (-0.092 to 0.012)

0.177

Mononuclear cell antrum

-0.052 (-0.067 to -0.037)

-0.036 (-0.082 to 0.009)

0.957

corpus

-0.064 (-0.084 to -0.045)

-0.051 (-0.088 to -0.013)

0.656

Neutrophil antrum

-0.028 (-0.043 to -0.012)

-0.060 (-0.102 to -0.018)

0.682

corpus

-0.042 (-0.063 to -0.022)

-0.070 (-0.110 to -0.030)

0.777

H. pylori   antrum

-0.032 (-0.049 to -0.014)

-0.067 (-0.128 to -0.006)

0.107

corpus

-0.038 (-0.057 to -0.019)

-0.044 (-0.078 to -0.010)

0.631

Abbreviations: PPIs, proton pump inhibitors.

Bold, statistical significance.

Table 4. Subgroup analysis of duration of H2RAs use of changes in the updated Sydney system score over time.

H2RAs   long term use (N=16)

H2RAs   short term use (N = 17)

P   value

Sydney   system factor

Score   coefficient (point), per year (95%CI)

Atrophy antrum

-0.037 (-0.117 to 0.044)

0.045 (-0.056 to 0.147)

0.382

corpus

-0.034 (-0.067 to -0.001)

0.005 (-0.056 to 0.066)

0.108

Metaplasia antrum

-0.053 (-0.100 to -0.0057)

-0.022 (-0.075 to 0.031)

0.296

corpus

-0.021 (-0.066 to 0.024)

-0.008 (-0.035 to 0.019)

0.350

Mononuclear cell antrum

-0.039 (-0.085 to 0.007)

-0.078 (-0.135 to -0.022)

0.567

corpus

-0.026 (-0.070 to 0.017)

-0.051 (-0.102 to -0.001)

0.988

Neutrophil antrum

-0.034 (-0.076 to 0.009)

-0.066 (-0.129 to -0.002)

0.681

corpus

-0.026 (-0.061 to 0.009)

-0.063 (-0.124 to -0.003)

0.692

H. pylori   antrum

-0.023 (-0.057 to 0.010)

-0.065 (-0.144 to 0.015)

0.443

corpus

-0.035 (-0.084 to 0.014)

-0.038 (-0.091 to 0.015)

0.633

Abbreviations: H2RAs, histamine 2 receptor antagonists.

2. Despite the hard work, the conclusions make this study less valuable. Because changes in IM were also noticed in considerable number of subjects, it should not be discriminated from atrophy. There are lot of recent studies showing that IM can be also improved after eradication. Therefore, the authors should better focus more on the duration of acid suppressants that diminished the effect of Hp eradication with regard to the improvement of atrophy and/or IM. ROC curve analysis might provide a solution for this.

Response: Thank you for this important comment. As shown above, our additional analysis showed a potential association between the duration of PPI use. However, ROC curve analysis showed a low AUC because of small sample size (attached file).

We are a bit confused by the reviewer’s dogmatic opinion that intestinal metaplasia and atrophy must be equally improved after eradication. We noted and now cited several recent studies showing the potential improvement of intestinal metaplasia after eradication, but regardless of these reports, this issue remains far apart from such definite conclusion. In a randomized controlled trial of H. pylori eradication, atrophy was improved in corpus greater curvature in eradicated patients, while intestinal metaplasia was not improved in corpus greater curvature [N Engl J Med. 2018;378:1085-1095]. In another recent observational study, atrophy was improved in both antrum and corpus, whereas intestinal metaplasia was not improved in antrum, and the improvement of atrophy and intestinal metaplasia was found only in females but not in males [Helicobacter. 2019 Mar 28: e12579]. The prospective study by Hwang et al may provide the most favorable results for the reviewer, showing a significant decrease of atrophy and intestinal metaplasia grade after eradication both in the corpus and antrum [AP&T, 2017]. Interestingly, the atrophy/intestinal metaplasia grades were rapidly dropped down within 1-2 years after eradication in the Hwang study, which was not observed in other previous studies. In any case, there are 2 meta-analysis studies which conclude that H. pylori eradication improves gastric atrophy but does not improve intestinal metaplasia [Rokkas et al, Helicobacter, 2007; Wang et al, Digestion, 2011], and another meta-analysis report which concludes that eradication correlates with improvement in intestinal metaplasia only in the antrum but not in the corpus [Kong et al, World J Gastroenterology, 2014].

Therefore, our results were not inconsistent with previous literatures, and we believe that our work is still valuable despite the unfavorable results for the reviewer. Nonetheless, due to the potential controversy on this point, we revised a few sentences in the conclusion in the Abstract.   

3. There are still several typos including line 59 Helicobacter eradication -> H. pylori eradication, line 156 quintiles, line 577 "stain" missing after immunohistochemical, etc.

Response: Thank you. We have corrected these errors.

4. Figure 1 (A) - Please consider removing the last box in the bottom. The readers may think that only 63 subjects completed this study among the 242 included subjects. Large number of drop-out rate might weaken this study.

Response: We have removed the last box in the bottom in Figure 1 (A).

5. Please minimize the numbers of keywords.

Response: We have minimized keywords as follows:

H. pylori eradication, atrophy, proton pump inhibitors (PPIs)”

Reviewer 2 Report

There are some minor word editing required: page 4, line 61, seems to be a more ...

Line 138: our previous works

Line 146: patients with...

The Figures are missing from the submitted (revised) manuscript, and should be presented so that the manuscript can be finally checked before acceptance.

Author Response

#Reviewer2

1. There are some minor word editing required:

page 4, line 61, seems to be a more ...

Line 138: our previous works

Line 146: patients with.

Response: Thank you. We have edited these words.

2. The Figures are missing from the submitted (revised) manuscript, and should be presented so that the manuscript can be finally checked before acceptance.

Response: We apologize again. There is a visibility problem in submitting web system. We have uploaded all manuscript and response to you as PDF file format. In addition, we contact an editor to send a visible manuscript file for you.